# A Fresh Look at Oral Chemolysis for Non-Symptomatic Kidney Stones—Comparative Research of Potassium Citrate and Allopurinol Combination—Is Treatment Possible Without Stone Analysis?

**DOI:** 10.3390/jcm14113970

**Published:** 2025-06-04

**Authors:** Alper Coşkun, Utku Can, Cengiz Çanakçı, Murat Can

**Affiliations:** Department of Urology, University of Health Sciences, Kartal Dr. Lutfi Kırdar City Hospital, 34865 Istanbul, Turkey; utkucan99@yahoo.com (U.C.); cengizcanakci@hotmail.com (C.Ç.); doctormrtcn@gmail.com (M.C.)

**Keywords:** allopurinol, lower calyx, oral chemolysis, potassium citrate, stone analysis

## Abstract

**Background/Objectives:** To compare the results of oral chemolysis of non-opaque and semi-opaque kidney stones using potassium citrate (PS) and allopurinol + potassium citrate (ALPS) prospectively without advanced metabolic analysis. **Methods**: Between 2020 and 2022, 59 patients aged 19–60 years with non-obstructive, semi-opaque, and non-opaque kidney stones of 8–13 mm in size detected by non-contrast tomography were investigated, and oral chemolysis (potassium citrate 10 mEq 2 × 2, allopurinol 10 mEq 2 × 2) was initiated. Patients were divided into two groups, potassium citrate (PS) and allopurinol + potassium citrate (ALPS), according to the treatment to be received. The essential characteristics of the groups, monthly laboratory values throughout the process, and the stone size measured by non-contrast tomography at the initial presentation and the stone size after a mean of 9–10 months of treatment was recorded and compared. **Results**: The inferior calyx was the most common location. There were no statistically significant differences between the groups with regard to age, sex, size, location or side. Before treatment, the mean stone size was 11.01 ± 2.27 mm in the potassium citrate group and 11.1 ± 2.03 mm in the other group. Following treatment, both groups changed significantly, but did not differ statistically (*p* < 0.001) (*p* = 0.281). The mean urine pH of all patients changed considerably after treatment (*p* < 0.001). **Conclusions**: Potassium citrate-based urinary alkalisation can be started for non-opaque and semi-opaque kidney stones without metabolic analysis if the urine pH is appropriate. The combination of allopurinol with potassium citrate has no therapeutic advantage.

## 1. Introduction

The primary treatment modalities for kidney stones are extracorporeal shock wave lithotripsy (SWL), retrograde intrarenal surgery (RIRS), and percutaneous nephrolithotomy (PCNL) [1,2]. Despite the fact that these procedures have been in use for a long time, developing a non-invasive treatment modality has always been a priority. Both ureteral stenting and chemolysis via percutaneous nephrostomy were attempted many years ago, but because minimally invasive methods are always on the agenda, these methods are no longer used [3].

Urinary alkalisation is now the most commonly chosen non-invasive treatment for urinary tract stone disease. Potassium citrate (PS), sodium bicarbonate, and magnesium bicarbonate have been the most popular molecules in this treatment [3]. In today’s urological practice, the most commonly used oral chemolysis molecule in cases of urinary calculi with hypocitraturia, hypercalciuria, hyperoxaluria, and hyperuricosuria is PS. It has been suggested that this molecule can be used without urometabolic analyses due to its wide range of indications [4,5]. Aside from the molecules used in oral chemolysis, allopurinol, a xanthine oxidase inhibitor, has long been used in the treatment of uric acid stones. Allopurinol has been shown to protect against uric acid stones and to reduce the size of the stones [6,7].

The status of follow-up and invasive treatment of non-symptomatic kidney stones less than 1 cm in diameter is still controversial, and metabolic analysis is not always possible. Given this situation, oral chemolysis comes to the fore in practice, and potassium citrate is the most commonly used molecule in the routine. However, it is unclear whether allopurinol provides any superiority in supportive treatment for these patients. Considering this situation, we started potassium citrate (PS) and allopurinol + potassium citrate (ALPS) treatment in some of the patients with a urine pH below seven who were admitted to our outpatient clinic and who were found to have kidney stones with a mean size of 10 mm on imaging and aimed to compare the effect of these treatments on stone size.

## 2. Material and Methods

### 2.1. Patients

Between March 2020 and February 2022, patients admitted to Kartal Dr. Lütfi Kırdar City Hospital’s urology outpatient clinic with incidental kidney stones underwent non-contrast tomography for further diagnosis. A total of 88 patients aged 19–60 years with a mean Hounsfeld Unit (HU) below 600 and with non-obstructive, semi-opaque and non-opaque 8–13 mm kidney stones were included in the study. Five of these patients were excluded because of comorbidities, four because of a low glomerular filtration rate (GFR), two because of solitary kidney and eight because of treatment refusal. Informed consent forms were completed from the remaining 69 patients, and the patients were divided into two groups: potassium citrate (PS) (n:33) and allopurinol+potassium citrate (ALPS) (n:36). PS was started at a dose of 10 mEq 2 × 2 and allopurinol 300 mg 1 × 1. In order to avoid bias in the selection and grouping of patients, no other criteria were determined except for stone size, non/semiopaque calculus and HU below 600. Age, gender, stone size (mm), location and side (right/left), HU, stone skin distance, body mass index (BMI), duration of treatment, urine pH, urea and creatinine values before and after medication, 24-h urine analysis, and whether there was spontaneous passage during treatment were recorded. Six patients from the PS group and four from the ALPS group were excluded from the study due to insufficient follow-up. Thus, 59 patients were included in the study.

Patients were invited to monthly controls throughout their treatment, and satisfaction with treatment, symptomatic status, and spontaneous passage were questioned. Participants in the ALPS group were also informed that they should be fed a purine-poor diet. In addition, every month, complete urinalysis, urea, and creatinine levels were measured. Treatment was typically continued for 9–10 months. Non-contrast tomography was used to determine the size of the stone after treatment.

### 2.2. Statistical Analysis

All qualitative and quantitative data were transferred to SPSS 22.0 (IBM^®^ SPSS^®^ Statistics V22.0, 2013, USA). Statistical results were obtained by comparing the groups within and between each other. An independent Samples *t*-test, Paired sample *t*-test, and Pearson Chi-Square, Mann–Whitney U, Wilcoxon Signed Ranks test were used for statistical analysis, and *p* < 0.05 was considered statistically significant.

## 3. Results

A total of 25 of the 59 patients were female, with 25 in the PS group and 34 in the ALPS group. The average age was 45 years, and 31 stones were in the right kidney. No patients had bilateral kidney stones. When individuals were classified based on the stone’s calyceal localisation, the lower calyx had the most cases (33). Of all the stones, 39% were semi-opaque, while 61% were non-opaque. The groups did not differ significantly in terms of age, stone size, Hounsfield unit (HU) values, or stone opacity (Table 1).

In the post-treatment controls of the patients, a significant change was observed in the stone size, and the mean urine pH was measured monthly (*p* < 0.001). However, no patient was found to have a urine pH above 7. Before treatment, the mean blood urea nitrogen (BUN) value was 25.5, while the mean urea value measured monthly was 27.6 (*p* = 0.026). There was no significant difference in creatinine and glomerular filtration rate (GFR) levels before and after treatment (Table 2). Both groups experienced a substantial decrease in stone size following treatment; however, there was no significant difference in stone size change when the groups were compared (*p* = 0.281) (Table 2).

## 4. Discussion

Urolithiasis is one of the most often diagnosed conditions in urology, with a worldwide prevalence ranging from 5 to 10%. The main determinants in the surgical and medical treatment of kidney stones are the size, location, obstruction status, and type of stone [1]. Surgical modalities are independent of the biochemical structure of the stone; however, the success of treatment in chemolysis is directly related to urine pH and stone analysis. Urinary alkalisation is the purpose of treatment and has been used for a long time. The best-known chemolysis has been described for uric acid stones [8,9,10,11]. The American Urological Association and European Association of Urology guidelines recommend offering potassium citrate (PS) therapy to patients with recurrent calcium stones and low (or relatively low) urinary citrate levels [1,12]. With this knowledge, it has been proposed that HU parameters will also be useful in predicting stone composition when selecting patients for oral chemolysis [13].

Chemolysis without stone analysis has always been contentious. In a recent study by Tsaturyan et al., potassium citrate, sodium bicarbonate, and magnesium bicarbonate were started in patients thought to have uric acid stones, and significant results were obtained [14]. Diri et al. also reported that urine pH measurement and X-ray features could provide information about the metabolic type of the stone, and that, for possible uric acid stones, oral hemolysin could be started [15].

It has long been debated whether allopurinol, a xanthine oxidase inhibitor used in the medical treatment of uric acid stones, has an effect on calcium-dominant kidney stones [6,16,17]. Generally, it has been supported that it prevents calcium oxalate stone formation secondary to hyperuricosuria [18,19]. With this assumption, we aimed to predict whether the combined use of PS/ALPS makes a difference in treatment by including non/semi-opaque kidney stone patients in our study. We should note that our results provide a preliminary conclusion that the addition of allopurinol is not significantly superior to standard treatment. The significant reduction in stone size in both treatments is attributed primarily to urinary alkalisation caused by the use of potassium citrate.

Another fundamental test in the metabolic investigation of kidney stones is the 24-h urine specimen. In general, 24-h urine supersaturation levels are suggested as the standard for forecasting the likelihood of stone development, particularly of calcium-dominant stones [1,12]. However, in a recent survey, it was reported that large centers dealing with stones routinely perform 24-h urine analysis, but they do not agree that this analysis is the best method, and there is uncertainty about how urine analysis should be performed and how the data should be interpreted. In addition, it has recently come to the fore that urine analysis at certain times of the day may be more informative than 24-h urine analysis [20,21]. One of the points of our study most open to criticism is that the 24-h urine analysis was performed only once before treatment and that the monthly controls of the patients were performed with a complete urinalysis instead of 24-h urine analysis. It is known that a single 24-h sample may not be sufficient before metabolic therapy [22]. However, we have to emphasise that there are problems with patient compliance when it comes to the collection of 24-h urine samples.

The small number of patients is another limitation of our study. We attribute this to the fact that the number of patients presenting with non- or semi-opaque kidney stones was generally insufficient, and our patients preferred a more definitive treatment. Another criticism could be directed towards the fact that the average length of treatment was 9–10 months. Our opinion on this subject is that more significant stone-free rates can be obtained with more prolonged treatment.

## 5. Conclusions

For non-opaque or semi-opaque non-obstructive kidney stones, oral chemolysis based on urine alkalisation is a safe and efficient treatment that, in the right individuals, can be administered without the need for stone analysis. The use of allopurinol in combination with potassium citrate is not significantly superior. Different results can be obtained with prospective and multicentric long-term studies involving more patients.

## Figures and Tables

**Table 1 jcm-14-03970-t001:** Characteristics of the groups and their comparison with each other.

All Patients (n: 59)		Mean ± SD	PS (n: 27)	ALPS (n: 32)	*p*
Age (years)		45.5 ± 10.7	41 ± 10.1	49 ± 10.1	0.936 **
Gender					0.266 *
	*Female*	n: 25	10	15	
	*Male*	n: 34	17	17	
BMI (kg/m^2^)		30.07 ± 3.9	28.1 ± 2.15	31.6 ± 4.21	0.009 **
Stone location					
*Upper calyx*		n:0			
*Middle calyx*		n:8	3	5	
*Renal pelvis*		n:18	7	11	
*Lower calyx*		n:33	16	17	0.894 *
Side of the stone					
*Right*		n:31	14	17	
*Left*		n:28	13	15	0.456 *
HU		519.35 ± 95.8	544.5 ± 72.2	500.2 ± 108.1	0.120 **
Opacity status					
*Non-opaque*		36 (61.0)	15	21	
*Semi opaque*		23 (38.9)	11	12	0.509 *
Stone skin distance (mm)		10.22 ± 1.89	9.67 ± 1.9	10.65 ± 1.79	0.551 **
Duration of treatment (month)		9.54 ± 2.2	8.75 ± 1.86	10.1 ± 2.34	0.033 **
Pre-treatment 24 h urine collection (mg/24 hrs)					
*Creatinine (mg)*		472.6 ± 57.1	475.6 ± 52.5	470.3 ± 61.4	0.790 ***
*Uric acid (mg)*		446.85 ± 102.4	463.5 ± 79.3	434 ± 117.1	0.549 ***
*Calcium (mg)*		172.6 ± 42	170.6 ± 50.7	174.1 ± 44.8	0.807 **
*Citrate (mg)*		552 ± 130	531.1 ± 141.4	568.1 ± 122.7	0.347 **
*Oksalat (mg)*		29.09 ± 5.4	28.6 ± 5.08	29.4 ± 5.72	0.797 ***
*Sodium (mEq)*		134.1 ± 17.2	134.2 ± 17	134.04 ± 17.7	0.981 **
*Potassium (mEq)*		35.61 ± 10.7	35.5 ± 11.5	35.7 ± 10.3	0.934 **
*Phosphate (mg)*		552.7 ± 130.8	582.4 ± 152.3	529.9 ± 109.1	0.362 ***

* Chi-Square, ** Independent Samples *t*-test, *** Mann–Whitney U Test. PS: potassium citrate, ALPS: allopurinol + potassium citrate, BMI: body mass ındex, BUN: Blood Urea Nitrogen, IQR: Interquartile range, HU: Hounsfield units.

**Table 2 jcm-14-03970-t002:** Comparison of all patients before and after oral chemolysis.

All Patients (n: 59)		PS (n: 27)	ALPS (n: 32)	*p*
Stone size before treatment (mm)	11.08 ± 2.11	11.01 ± 2.27	11.1 ± 2.03	0.816 **
Stone size after treatment (mm)	8.43 ± 3.42	9.34 ± 2.27	7.73 ± 3.99	0.281 ***
*p* < 0.001 *		<0.001 *	<0.001 *	
Urea (BUN) before treatment	25.51 ± 6.26	24.68 ± 5.51	26.15 ± 6.81	0.435 **
Urea(BUN) after treatment	27.6 ± 5.96	26.44 ± 4.0	28.5 ± 7.05	0.235 **
*p* < 0.026 *		0.259 *	0.049 *	
Pre-treatment creatinine	0.76 ± 0.15	0.74 ± 0.13	0.78	0.351 **
Post-treatment creatinine	0.82 ± 0.18	0.80 ± 0.17	0.83	0.654 **
*p* < 0.014 *		0.058 *	0.123 *	
Pre-treatment GFR	100.65 ± 15.05	103.39 ± 15.26	98.54	0.297 ***
Post-treatment GFR	96.2 ± 19.84	99.1 ± 20.72	94.06	0.277 ***
*p* < 0.010 ****		0.040 *	0.01 *	
Pre-treatment urine ph	5.89 ± 0.37	5.97 ± 0.37	5.83 ± 0.36	0.292 ***
Post-treatment urine ph	6.27 ± 0.29	6.37 ± 0.31	6.20 ± 0.26	0.189 ***
*p* < 0.001 ****		<0.001 *	<0.001 *	

* Paired sample *t*-test, ** Independent Samples *t*-test, *** Mann–Whitney U Test, **** Wilcoxon Signed Ranks Test, PS: potassium citrate, ALPS: allopurinol + potassium citrate, GFR: Glomerular filtration rate.

## Data Availability

Dataset available on request from the authors.

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
