# Peer review of "A Fresh Look at Oral Chemolysis for Non-Symptomatic Kidney Stones—Comparative Research of Potassium Citrate and Allopurinol Combination—Is Treatment Possible Without Stone Analysis?"

_jcm, 2025, doi:10.3390/jcm14113970_

Round 1

Reviewer 1 Report

Comments and Suggestions for Authors

The main question is to compare the results of oral chemolysis of non-opaque and semi-opaque kidney stones using potassium citrate (PS) and allopurinol potassium citrate (ALPS) prospectively without advanced metabolic analysis.

The topic is original or relevant to the field.

The results provide a preliminary conclusion that the addition of allopurinol is not significantly superior to standard treatment. The significant reduction in stone size in both treatments is attributed primarily to urinary alkalization caused by the use of potassium citrate. 

well addressed methodology.

Conclusions are consistent with the evidence.

References are appropriate.

No further comments on the tables and figures.

Author Response

Response to Reviewer

Review 1.

  1. The main question is to compare the results of oral chemolysis of non-opaque and semi-opaque kidney stones using potassium citrate (PS) and allopurinol potassium citrate (ALPS) prospectively without advanced metabolic analysis.:

Response:

Dear reviewer;

The description you use exactly characterises our article. For this reason, I would like to thank you.

  1. The topic is original or relevant to the field.
  • Response:

Dear reviewer;

Thanks to with my deepest respect.

  1. The results provide a preliminary conclusion that the addition of allopurinol is not significantly superior to standard treatment. The significant reduction in stone size in both treatments is attributed primarily to urinary alkalization caused by the use of potassium citrate.
  • Response:

Your statement is exactly the conclusion of our article

Kind regards

  1. Well addressed methodology

Response:

Thanks for your comments

Kind regards

  1. Conclusions are consistent with the evidence

Response:

Thanks for your comments

Kind regards

  1. References are appropriate.

Response:

Thanks for your comments

Kind regards

  1. No further comments on the tables and figures. ‘’ Tablo ve ÅŸekillerle ilgili baÅŸka yorum yok.’’

Response:

Thanks for your comments

Kind regards

Reviewer 2 Report

Comments and Suggestions for Authors
  • There is no control group in this experiment which ignores placebo bias
  • How were the patients randomized 
  • How was the follow up ensured, what was the complaince rate in either arm? These questions need to be answered 
  • Additionally language needs to improved, for example 

    "Eight of these patients were excluded because of treatment refusal due to

    recurrent pain, five comorbidities, four low glomerular filtration rate (GFR), and two

    solitary kidney"

    — This statement needs clarity in writing. Should be reformatted.

Comments on the Quality of English Language

- Language needs to be worked on. 

Author Response

Response to Reviewer

Review 2.

  1. There is no control group in this experiment which ignores placebo bias

Response:

Dear reviewer;

First and foremost, I'd like to thank you for your valuable comments. There are two groups in our article and the people in each group are kidney stone patients. For this reason, it is not possible to include a placebo group in terms of study design.

Kind regards

  1. How were the patients randomized?

Response:

Dear reviewer;

Our patients were randomised according to the type of treatment they would receive for kidney stones.

Kind regards

  1. How was the follow up ensured, what was the complaince rate in either arm? These questions need to be answered.

Response:

Dear reviewer;

As I mentioned in the title of my article; patients with asymptomatic kidney stones were included in the study. In the Material and Methods 2.1 Patients section, deleted sentences between paragraphs 66 and 74 are crossed out in red and newly added sentences are indicated in yellow. Again, between 89th and 94th sentences in the material and method section, it is stated how the patients were followed up.

Kind regards

  1. Additionally language needs to improved, for example "Eight of these patients were excluded because of treatment refusal due to recurrent pain, five comorbidities, four low glomerular filtration rate (GFR), and two solitary kidney" — This statement needs clarity in writing. Should be reformatted.

Response:

Dear review

In general, the necessary arrangements have already been made in terms of language; I should state that no additional arrangements have been made in its current form. The sentence you mentioned; has been changed. The previous sentence is red and crossed out. The new sentence is in yellow between lines 76-78.

Reviewer 3 Report

Comments and Suggestions for Authors

I would like to receive answers to the following comments
1 Was the diet taken into account in the studied groups
2 Were there plant components in the food products that ensured the alkalinity of the food consumed
3 Opaque and transparent stones have different chemical compositions. How can it be explained THAT THE PROPOSED PREPARATIONS do not reliably affect the genesis of their development

Author Response

Response to Reviewer

Review 3

I would like to receive answers to the following comments;

  1. Was the diet taken into account in the studied groups

Response;

Dear reviewer;

First of all, I would like to express my gratitude for your valuable comments. We did not make any dietary recommendations to the groups except for abundant hydration. I should mention that we did not make additional dietary recommendations because we wanted to compare the results of oral chemolysis.

Kind regards

  1. Were there plant components in the food products that ensured the alkalinity of the food consumed

Response;

Dear reviewer;

There were not.

Kind regards

  1. Opaque and transparent stones have different chemical compositions. How can it be explained THAT THE PROPOSED PREPARATIONS do not reliably affect the genesis of their development.

Response:

            Dear reviewer;

Our patient population consists of nonopaque and semiopaque stone patients. The treatment modalities we initiate for our patients are determined according to the parameters in the 24-hour urine analysis and complete urinalysis findings. For this reason, we attribute the reliability of the treatments to the fact that the drugs used are known substances in the metabolic treatment of kidney stones.

Kind regards

Round 2

Reviewer 2 Report

Comments and Suggestions for Authors

Please have a placebo arm to make the study more accurate. 

Author Response

Response to Reviewer

Review 2.2.

First of all, I would like to thank you again for your valuable comments. answered your comments in items.

  1. Is the research design appropriate? Can be improved

Response: I read your feedback on the design of our study. I declare that no further steps can be taken in terms of additional editing of the overall of all  authors.

Kind regards

  1. Are the methods adequately described? Can be improved

Response: The necessary adjustments are indicated in the materials and methods section and are marked in yellow and red. No additional edits were made.

Kind regards

  1. Are the results clearly presented? Can be improved

Response: Necessary arrangements were made in the Results section. The sentences between 103-107 have been changed and highlighted in yellow.

Kind regards

  1. Are the conclusions supported by the results? Must be improved

Response: We believe that our results are clearly presented in tables. Unfortunately, no additional adjustments could be made.

Kind regards

  1. Are all figures and tables clear and well-presented? Can be improved

Response: As stated in the previous item; ‘we are of the opinion that our results are clearly presented in tables’.

  1. Comments and Suggestions for Authors. Please have a placebo arm to make the study more accurate.

Response: There are two groups in our study and the patients in both groups are individuals with a confirmed diagnosis of kidney stones. The additional group in this version of our study changes our article completely. This gives an idea for the next study. Thank you for this sensitivity.

Kind regards